# Prevalence of HIV testing uptake among the never-married young men (15–24) in sub-Saharan Africa: An analysis of demographic and health survey data (2015–2020)

Emmanuel Musonda[1]*, Million Phiri[1,2], Liness Shasha[1], Chiti Bwalya[3], Shuko Musemangezhi[4], Sage Marie Consolatrice Ishimwe[5], Chester Kalinda[4]

1 Department of Population Studies, School of Humanities and Social Sciences, University of Zambia, Lusaka, Zambia, 2 Demography and Population Studies Programme, Schools of Public Health and Social Sciences, University of the Witwatersrand, Johannesburg, South Africa, 3 Maryland Global Initiatives Corporation (MGIC), Lusaka, Zambia, 4 Bill and Joyce Cummings Institute of Global Health, University of Global Health Equity, Kigali, Rwanda, 5 Institute of Global Health Equity Research (IGHER), University of Global Health Equity, Kigali, Rwanda

* musondaemmanuel11@gmail.com

## Abstract

### Background

In sub-Saharan Africa, HIV and AIDS remain a major public health concern among adolescents and young men. HIV testing is the first critical step for linking infected individuals to HIV treatment and prevention. However, HIV-testing uptake among sexually active young men remains low in the region. This study was conducted to assess the HIV testing rates among unmarried young men in sub-Saharan Africa.

### Methods

Using data from the most recent country Demographic and Health Surveys (DHS) conducted between January 1, 2015, and December 31, 2020, in 18 sub-Saharan African countries, an Inverse Heterogeneity model (IVhet) using MetaXL software was used to estimate country, regional and sub-regional pooled estimates of HIV testing uptake among sexually active unmarried young men in sub-Saharan Africa. Furthermore, multivariable binary logistic regression was conducted to examine the factors associated with HIV testing uptake among unmarried young men.

### Results

The overall pooled prevalence estimate of HIV testing uptake among sexually active unmarried young men in sub-Saharan Africa was 33.0% (95% CI: 21–45, I2 = 99%, p <0.001). There was variation in the prevalence across countries ranging from 7% (95% CI: 5–9) in Guinea to 77% (95% CI: 74–80) in Cameroon. Central Africa had the highest prevalence of HIV testing among unmarried young men, at 47% (95% CI:0–100) while West Africa had the lowest prevalence at 11% (95% CI:2–23). Results further show that young men aged 15–19

**Data Availability Statement:** Data used in our study is publicly available at DHS program website (https://dhsprogram.com/).

**Funding:** The author(s) received no specific funding for this work.

**Competing interests:** The authors have declared that no competing interests exist.

**Abbreviations:** AIDS, Acquired Immune Deficiency Syndrome; CI, Confidence Interval; DHS, Demographic and Health Survey; EA, Enumeration Area; HIV, Human Immunodeficiency Virus; SDG, Sustainable Development Goal; SRH, Sexual Reproductive Health; SSA, Sub-Saharan Africa; UNFPA, United Nations Population Fund; UNICEF, United Nations Children Fund; USAID, United States Aid for International Development; WHO, World Health Organisation; ZDHS, Zambia Demographic and Health Survey.

(aOR = 0.59, 95% CI 0.52–0.66) were less likely to test for HIV. Young men who spent 8 to 12 years in school (aOR = 3.26 95% CI 2.21–4.79) or 13 years and above (aOR = 3.56 95% CI 2.35–5.37) had increased odds of undertaking an HIV test.

## Conclusion

The prevalence of HIV testing among sexually active unmarried young men remains low in sub-Saharan Africa. Therefore, the results suggest that health policymakers should consider re-evaluating the current HIV prevention policies and programmes with the view of redesigning the present HIV testing campaigns to enhance the uptake among young people.

## Introduction

Emerging evidence suggests that globally, around 37.7 million people are living with HIV and of these, 71% of cases are from sub-Saharan Africa (SSA) [1]. Global efforts into achieving the 95-95-95 goals and United Nations Children's Fund (UNICEF) and the Joint United Nations Programme on HIV/AIDS (UNAIDS) initiatives have led to increased availability of HIV testing and treatment [2]. Although knowledge of individual serostatus is key in linking infected persons to lifesaving antiretroviral therapy (ART) with a potential reduction in HIV spread, the lag in the number of people testing and knowing their status despite living with HIV remains low; thus increasing the risks of early antiretroviral therapy (ART) initiation [3]. Delayed HIV diagnosis ultimately impacts the initiation of ART leading to severe outcomes, increased morbidity and mortality, and increased HIV transmission [4–7]. Therefore, understanding the prevalence and factors associated with HIV testing among sexually active unmarried young men would be critical in achieving the 95-95-95 goals and all its initiatives [8, 9].

According to the UNAIDS and UNICEF reports of 2016, SSA has about 2.1 million adolescents living with HIV [10, 11]. Even with the significant progress in voluntary HIV counselling, testing and prevention measures, as well as the introduction of rapid diagnostic tests and self-test delivered through community and home based-strategies, the testing rates among young men remain low across SSA countries [12].

Several studies have highlighted factors that negatively affect the uptake of HIV testing among married or unmarried young men [8, 13–18]. At the proximal level, the perception of a low risk of HIV infection, the emotional burden of dealing with a positive result, the absence of support from family and friends, and daily mobility linked to livelihood options, social recreation, and daily mobility to and from school act as barriers [1, 19, 20]. At the distal level, legal barriers such as the age of consent and parental consent laws, health system barriers such as stigma, perceived lack of confidentiality and fear of disrespect by health staff discourage young men from accessing HIV testing services [19].

Significant progress has made towards curbing the HIV pandemic in SSA. However, HIV testing among youths remains a challenge [12]. Although strategies such as HIV self-testing have been developed to reinforce current efforts, several studies conducted in SSA have documented HIV testing among adults and pregnant women, while gaps in the prevalence of HIV testing among adolescents and youths remain unknown [19]. This study was therefore conducted to establish the prevalence of recent HIV testing among sexually active never-married young people 15–24. This information would be very vital in providing decision-makers with evidence to increase HIV-testing coverage, targeting young men [21]. Furthermore, HIV testing data on country and regional disparities would be important to guide the implementation

of best practices on sexual and reproductive health [22]. In this study, we applied a meta-analysis to determine the pooled prevalence as well as examine the country and sub-regional heterogeneity of recent HIV testing among unmarried young men 15–24 years in SSA using recent nationally representative data.

## Methods and data

### Data sources

The data used in this study was extracted from the most recent Demographic and Health Survey (DHS) conducted between 2015 and 2020 in 18 SSA countries. DHS datasets are readily available to the public on the DHS website, https://dhsprogram.com/. Each survey is periodical and population-based, comprising multi-stage stratified samples between 5000 and 30000 people. As a way of enabling comparisons among regions and countries, DHS gathers data using a standardised tool comprising household, women's, men's and biomarker questionnaires. In addition, the survey employs a multi-stage stratified design with probabilistic sampling, where each household has an equal chance of selection. Every survey was stratified by rural and urban status and country-specific geographic or administrative regions, such as provinces or regions. An elaborated sampling and data collection plan is available from the survey's final reports [23].

### Study countries

There are 48 countries in SSA. Of the 48 counties, 43 have conducted at least one DHS. Thirty-nine (39) countries had accessible DHS datasets. Of the 39 countries that had accessible DHS data, 18 countries conducted the most recent DHS between 2015 and 2020. Thus, all 18 countries with accessible DHS datasets conducted between 2015 and 2020 were included in the present study. Because some countries did not have latest DHS data, therefore, DHSs conducted between 2015 and 2020 were considered to provide a clear picture of the prevalence of recent HIV testing among sexually active never-married young men in SSA. The sub-regional classification of countries in SSA is based on the United Nations (UN) geo-scheme classification.

### Data extraction

The DHS datasets were sourced from the DHS program website, encompassing 18 countries in SSA from 2015 to 2020. The DHS dataset we used for the study contains only data for men aged 15–59 years (MR recode). Our decision to focus our study on unmarried young men was informed by existing literature which shows that unmarried young men are more likely to engage in behaviours that put them at higher risk of HIV infection, such as multiple sexual partners, unprotected sex, or substance use [24–27]. Assessing HIV testing rates among young men allows healthcare providers and policymakers to identify gaps in testing coverage and develop targeted interventions to increase testing rates. The extracted information from the country-level datasets comprised the name of the country, year of DHS implementation, weighted sample of the sexually active never-married men aged 15–24 years, counts of sexually active men who underwent an HIV test in the past 12 months before the survey, and sub-region (Table 1).

### Study measures

**Dependent variable.** The dependent variable of interest in this study is testing for HIV in the last 12 months before the survey. All sexually active men who were interviewed in the DHS

**Table 1. Distribution of sexually active-unmarried young men (15–24) who tested for HIV in the last 12 months preceding the DHS in SSA countries (2015–2020).**

| Country | DHS Year | Weighted sample | Number of sexually active unmarried men who had recent HIV test | Percentage who had an HIV test and received the results | Region |
|---|---|---|---|---|---|
| Angola | 2015 | 1,574 | 264 | 14.6 | Southern Africa |
| Benin | 2017 | 1093 | 91 | 7.9 | West Africa |
| Burundi | 2016 | 559 | 136 | 23.6 | East Africa |
| Cameroon | 2018 | 645 | 497 | 77.5 | Central Africa |
| Chad | 2015 | 527 | 69 | 13.8 | Central Africa |
| Ethiopia | 2016 | 788 | 326 | 38.8 | East Africa |
| Gambia | 2019 | 78 | 33 | 42.6 | West Africa |
| Guinea | 2018 | 595 | 39 | 7.2 | West Africa |
| Liberia | 2019 | 752 | 78 | 11.7 | West Africa |
| Malawi | 2015 | 1598 | 661 | 40.2 | Southern Africa |
| Mali | 2018 | 412 | 35 | 7.3 | West Africa |
| Rwanda | 2019 | 492 | 292 | 43.4 | East Africa |
| Senegal | 2019 | 783 | 74 | 7.2 | West Africa |
| Sierra Leone | 2019 | 194 | 141 | 72.9 | West Africa |
| South Africa | 2016 | 873 | 420 | 49.3 | Southern Africa |
| Uganda | 2016 | 959 | 488 | 49.9 | East Africa |
| Zambia | 2018 | 2564 | 1370 | 56.0 | Southern Africa |
| Zimbabwe | 2015 | 1323 | 517 | 39.1 | Southern Africa |

were asked a question on whether they had undertaken an HIV test in the past 12 months prior to the survey. This variable was categorised as a yes or no response. The analysis was restricted to sexually active unmarried young men aged 15–24 years.

**Independent variables.** Based on the review of existing literature on HIV and AIDS in SSA and elsewhere [13, 18, 28–31], the study identified correlates at individual and household levels that could be potentially associated with HIV testing among young men. These variables were classified as socio-economic and demographic. The DHS reference materials and data collection were used to identify the independent variables of interest presented in this section. The following independent variables were included in the study analysis: the age of young men categorised as 15–19, 20–24; total years spent in education was coded as Less than 1 year, 1–7 years, 8–12 years, 13 or more years; wealth index in DHS is usually captured as 5 response categories: poorest, poor, middle, rich, richest. For this study's analysis, we recoded this variable with the following categorisation: poor, middle and rich; occupation status was categorised as employed, unemployed; age at first sex (15 years, 15 or more); circumcision status (no, yes); number of lifetime sex partners (1 partner, 2–3 partners, 4 or more partners); time since last sexual intercourse in days (less than 30 days, 30 or more days).

## Statistical analysis

Statistical analysis was conducted on the pooled dataset comprising 18 DHSs. MetaXL (version 5.3, EpiGear International Pty Ltd, QLD, Australia) was used to perform the descriptive and statistical analysis. The overall prevalence was calculated and its associated 95% confidence interval (CI) for the pooled recent HIV testing among the sexually active unmarried men aged

15–24. Country-specific HIV testing prevalence estimates were computed using the Inverse Heterogeneity (IVhet) model to produce the HIV testing estimates. The IVhet model maintains a correct coverage probability at a lower detected variance. Sub regional pooled prevalence was estimated for (West Africa, Central Africa, East Africa and Southern Africa). The estimated prevalence for individual countries and pooled sub-region was displayed using the forest plot and its associated 95% confidence intervals (CI). The (I2)- statistic was used to quantitatively evaluate the heterogeneity, while the Luis Furuya-Kanamori (LFK) index of the Doi Plot was used to assess the publication bias. Furthermore, a multivariable binary logistic regression model was conducted on pooled data to examine the factors associated with the uptake of HIV testing among unmarried young men in SSA. Sample weights were equalised to give equal weights to each survey included in the analysis.

## Ethical approval

The study relied on secondary data sources. Permission to use DHS datasets was obtained from the DHS program. In the DHS data, there are no personal identifiers for survey participants. The original DHS Biomarker and survey protocols for respective countries were approved by the country's Ethical Review Bodies and the Research Ethics Review Board of the Center for Disease Control and Prevention (CDC) Atlanta. All DHS participants 18 or older were required to consent to interviews. For all participants aged 15 to 17, the DHS policy needed parental/guardian consent before requesting assent from legal minors. Data analysed in this study is available in the public domain (https://dhsprogram.com/).

## Results

### Study characteristics

A total of 18 SSA DHS datasets were analysed for this study (**Table 1**). From Central Africa, 2 (11.1%) countries were included (Cameroon, Chad), Eastern Africa had 4 (22.2%) countries included (Burundi, Ethiopia, Rwanda, Uganda) while Southern Africa had 5 (27.8%) countries included (Malawi, Zambia, Zimbabwe, Angola, South Africa), and West Africa had 7 (38.9%) countries included (Benin, Gambia, Liberia, Mali, Senegal, Sierra Leone, Guinea). Our study results show no publication bias because the Luis Furuya-Kanamori index of (-0.45) was within the symmetry range of -1 and +1 (**S1 Fig**).

### Overall pooled prevalence of HIV testing uptake

The pooled prevalence estimate (PPE) for recent HIV testing among unmarried young men from 18 SSA countries was 33.0% (95% CI: 21–45, I2 = 99%, *p<0.001*). There was variation in the prevalence across countries ranging from 7% (95% CI: 5–9) in Guinea to 77% (95% CI: 74–80) in Cameroon (**Fig 1**; **S1 Table**). Only 4 of the 18 countries included in the analysis had a prevalence of recent HIV testing among unmarried young men of 50% or above. These are Cameroon from Central Africa, Uganda from East Africa, Zambia from Southern Africa and Sierra Leone from West Africa (**Fig 1**).

### Prevalence of HIV testing uptake by sub-region

The pooled prevalence by sub-region showed that Central Africa had the highest prevalence, 47% (95% CI:0–100, $I^2$ = 100, *p<0.001*) of recent HIV testing among sexually active unmarried young men, whilst West Africa had the lowest prevalence, 11% (95% CI:2–23, $I^2$ = 99, p<0.001) (**Table 2**).

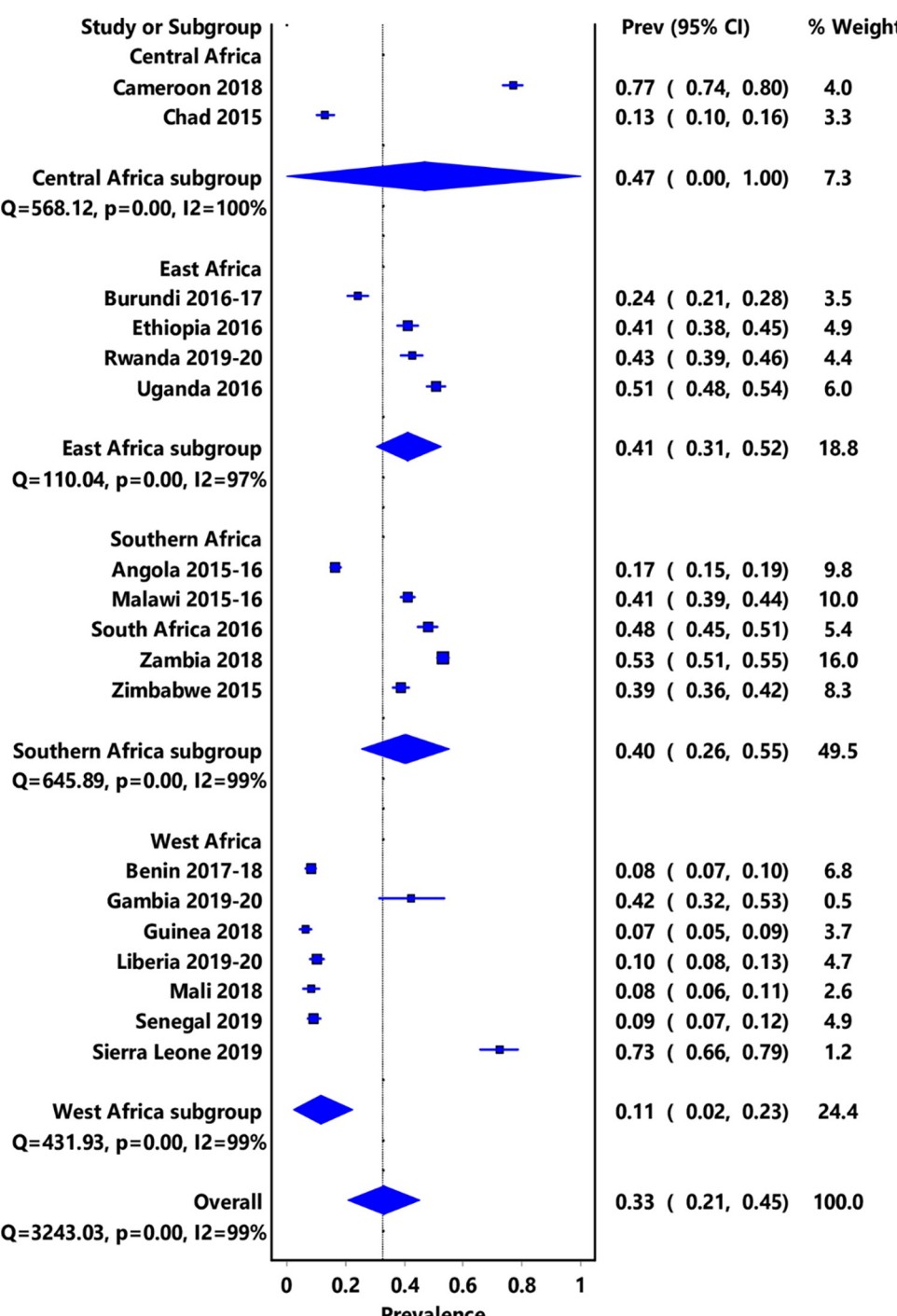

**Fig 1. Prevalence of HIV testing uptake among sexually active unmarried young men aged 15–24 in SSA countries.**

### Determinants of HIV testing uptake among young men

Table 3 presents results from the multivariable regression model showing an association between all independent variables and HIV testing uptake among unmarried young men. Results show that age, years spent in school, household wealth status, occupation, circumcision

**Table 2. Prevalence of HIV testing uptake among sexually active unmarried young men (15–24) by sub-region.**

| Region | Sample size | Prevalence (95%CI) | I2 | p-value |
|---|---|---|---|---|
| Central Africa | 1,172 | 0.47 (0.00, 1.00) | 100 | 0.000 |
| East Africa | 3,012 | 0.41 (0.31, 0.52) | 97 | 0.000 |
| Southern Africa | 8,744 | 0.40 (0.26, 0.55) | 99 | 0.000 |
| West Africa | 2,814 | 0.11 (0.02, 0.23) | 99 | 0.000 |

status, number of lifetime sex partners, and condom use during the last sex with the most recent partner were all factors associated with HIV testing uptake. Young men aged 15–19 (aOR = 0.59; 95% CI: 0.52–0.66) were less likely to test for HIV compared to those aged 20–24. Regarding time spent in education, young men who spent 8 to 9 years (aOR = 3.26; 95% CI: 2.21–4.79) and those who spent 13 years or above (aOR = 3.56; 95% CI: 2.35–5.37) were more likely to test for HIV compared to those who spent less than 1 year in school. Young men who belonged to poor households (aOR = 0.78; 95% CI: 0.67–0.90) were less likely to test for HIV compared to those who were from rich households. Young men who were employed (aOR = 1.14; 95% CI: 1.01–1.30) were more likely to test for HIV compared to the unemployed. Young men who were circumcised were less likely to test for HIV compared to those who were not circumcised (aOR = 0.66; 95% CI: 0.59–0.74).

## Discussion

This study was conducted using DHS data for 18 countries to determine the prevalence and examine the factors associated with HIV testing among the never-married young men aged 15–24 years in SSA. The study results show that the pooled prevalence of HIV testing among the never-married young men is still low, 32% (95% CI: 21,44). A study conducted by Asaolu and others in 2016 reported a prevalence of 23% among young people in SSA [14]. The results show that Cameroon had the highest proportion of HIV testing among never-married young men, 77% (95% CI: 74,80) while Guinea had the lowest proportion of HIV testing uptake among the never-married young men at 7% (95% CI: 5,9). A study conducted by Staveteig and others in 2013 reported that Malawi had the highest proportion of youths 20–24 years (84%) who tested for HIV and the least prevalence was reported in Chad at 1% [32].

HIV testing rates varied across sub-regions in SSA. Central Africa showed the highest prevalence, 47% whilst West Africa had the lowest prevalence at 11%. This result is partly similar to what was reported by a previous study conducted by Asaolu and others in 2016, which showed that Eastern Africa (78%) had the highest proportion of youths that tested for HIV and the least was Western Africa (31%) [14]. One reason for the variation in HIV testing across countries and regions could be variations in HIV policy and policy programme implementation across countries [33]. Additionally, differences in sociocultural values and beliefs could also contribute to variations in acceptance of HIV testing among young people [34–37]. For instance, Malawi has been particularly advanced in promoting the rapid initiation of antiretroviral therapy [38].

The information about the pooled prevalence of HIV testing among sexually unmarried young men in SSA is essential for designing HIV programmes, taking into consideration specificities that may be related to the age group [12]. The results from this current study show that the prevalence of HIV testing among young men remains low in SSA. This has the potential to increase the risk of not achieving the UNAIDS "95-95-95" initiative [20]. Therefore, there is a need to identify and scale up strategies that enhance HIV diagnosis and to have infected young people start treatment at much higher than current CD4 counts [39]. Despite the progress in terms of strategy and policy to improve access to HIV services, a huge disparity exists in terms

**Table 3. Adjusted odds ratios for the multivariable binary logistic regression of the association between independent variables and HIV testing uptake among sexually young unmarried young men aged 15–24 years in SSA countries.**

| Background Characteristics | (N = 10,122) | | |
|---|---|---|---|
| | Adjusted Odds Ratios | p-value | (95% CI) |
| **Age** | | | |
| 15–19 | 0.59 | p<0.001 | 0.52–0.66*** |
| 20–24 | 1 | | |
| **Total years of education** | | | |
| Less than 1 | 1 | | |
| 1–7 | 2.15 | p<0.001 | 1.48–3.13*** |
| 8–12 | 3.26 | p<0.001 | 2.21–4.79*** |
| 13+ | 3.56 | p<0.001 | 2.35–5.37*** |
| **Household wealth status** | | | |
| Poor | 0.78 | p<0.01 | 0.67–0.90** |
| Middle | 0.93 | p>0.05 | 0.81–1.08 |
| Rich | 1 | | |
| **Working status** | | | |
| Unemployed | 1 | | |
| Employed | 1.14 | p<0.05 | 1.01–1.30* |
| **Age at first sex** | | | |
| Less than15 | 1 | | |
| 15 or more | 1.15 | p>0.05 | 0.99–1.33 |
| **Paid for sex in last 12 months** | | | |
| No | 1 | | |
| Yes | 0.90 | p>0.05 | 0.76–1.06 |
| **Circumcision status** | | | |
| No | 1 | | |
| Yes | 0.66 | p<0.001 | 0.59–0.74*** |
| **Number of lifetime sex partners** | | | |
| **1** | 1 | | |
| **2–3** | 1.16 | p<0.05 | 1.01–1.33* |
| **4+** | 1.30 | p<0.01 | 1.10–1.52** |
| **Time since last sexual intercourse in days** | | | |
| Less than 30 days | 1 | | |
| 30 or more days | 0.63 | p>0.05 | 0.39–1.04 |
| Used condom during last sex with most recent partner | | | |
| **No** | 1 | | |
| **Yes** | 1.58 | p<0.05 | 1.41–1.77*** |
| **Sexual activity in past 4 weeks** | | | |
| No | **1** | | |
| Yes | 0.69 | p>0.05 | 0.42–1.13 |

*** = $p < 0.001$

** = $p < 0.01$

* = $p < 0.05$

of the prevalence of HIV testing among SSA countries exists. One way for the future global response to HIV is sustained donor funding accompanied by innovative financing and strategic redeployment of health system assets to ensure that the delivery of HIV services is resilient and sustainable [40].

Our study further found that age, number of years spent in school, wealth index, occupation, circumcision status, number of lifetime sex partners, and condom use were significantly associated with HIV testing among unmarried young men (15–24) in SSA. In this current analysis, young men aged 15–19 years were less likely to test for HIV. Studies conducted elsewhere support these findings [14, 35, 41]. One explanation for this could be the fact that many young people rarely receive thorough sexuality education and are less aware of the health hazards connected with HIV. This makes them less knowledgeable about the value of routine HIV testing and the advantages of early detection [42, 43].

Several studies have reported the association between education with HIV testing in SSA [44–46]. A similar finding from our study revealed that an increase in the number of years spent in school increases the likelihood of testing for HIV. This implies that education is essential for boosting knowledge and understanding of HIV transmission, prevention, and the value of routine testing. People with higher levels of education are more likely to have access to reliable information and comprehend the risks connected with HIV, which promotes a better feeling of personal responsibility for one's health [9, 47]. Second, socioeconomic status and education are frequently correlated, and this can facilitate access to healthcare services, including HIV testing. People who are more educated might have access to superior financial resources, health insurance coverage, and knowledge of nearby medical facilities, making it simpler for them to get tested for HIV [46, 48].

Young men belonging to poor households had lower odds of testing for HIV in SSA. This finding supports the conclusion of a previous study, which showed that belonging to poor households is associated with low HIV testing [8]. One explanation could be that people from low-income households frequently encounter obstacles while trying to get healthcare services, such as HIV testing. People may choose not to get tested for HIV because of a lack of money, a lack of health insurance, or a long commute to a hospital [32, 49].

This study has provided useful findings that have the potential to inform the strengthening of existing HIV prevention programmes tareting young people. Enhancing uptake of HIV testing among young people will play a vital role in reducing the risk of acquiring HIV among adolescents and young men hence optimising their health and well-being [50]. HIV testing uptake among young people can be improved by enhancing deferential HIV programming activities and encouraging self HIV-testing strategies that overcome obstacles such as stigmatisation and discrimination [30]. Furthermore, increasing awareness, acceptance, and support for age-specific sexuality health education campaigns need to be enhanced to improve HIV-testing uptake.

## Limitations of the study

The study has provided a comprehensive picture of the prevalence of HIV testing rates and factors associated with the uptake of HIV testing among sexually active young people in SSA. However, there are limitations to the study. First, not all SSA countries were analysed because most countries have DHS data collected before 2015. Therefore, the findings in this study would be generalised only to the group of countries considered in the analysis. Second, the surveys used in our analysis were not conducted during the same period, thus the pooled prevalence reported covers the period 2015 to 2020. Last, the DHS collects information for events that happened prior to the data collection exercise, therefore, data may suffer from recall bias. Despite these limitations, the DHS provides useful national-level health indicators which can inform the design of targeted public health policies and interventions aimed at reducing the risk of HIV infections among young people in SSA.

## Conclusion

The study has established that HIV testing uptake among sexually active unmarried young men in SSA is low. There are significant variations in HIV testing uptake across countries and sub-regions. Age, number of years spent in education, employment status, male circumcision status and number of sexual partners were associated with HIV testing uptake among young men. The study findings imply that investing in education sector to increase access among young people has the potential to increase the uptake of HIV testing. The findings may suggest the need for improved implementation of age-specific social behaviour change activities to enhance HIV testing uptake among sexually active young men. Building on past successful best practices, HIV policies and programmes and blending them with new strategies would be essential in increasing HIV testing uptake among young men. There is a need for further research to understand best practices regarding sexual and reproductive health and HIV prevention programming from better-performing countries.

## Supporting information

**S1 Fig. Doi plot show publication measure of bias for the studies included.**
(DOCX)

**S1 Table. Sensitivity analysis results.**
(DOCX)

## Acknowledgments

We appreciate the Demographic and Health Survey Program, ICF and other partners involved in the DHS program.

## Author Contributions

**Conceptualization:** Emmanuel Musonda, Million Phiri.

**Data curation:** Emmanuel Musonda, Liness Shasha.

**Formal analysis:** Emmanuel Musonda, Million Phiri.

**Methodology:** Chiti Bwalya.

**Project administration:** Emmanuel Musonda.

**Software:** Million Phiri, Sage Marie Consolatrice Ishimwe.

**Writing – original draft:** Million Phiri, Chester Kalinda.

**Writing – review & editing:** Emmanuel Musonda, Million Phiri, Liness Shasha, Chiti Bwalya, Shuko Musemangezhi, Sage Marie Consolatrice Ishimwe, Chester Kalinda.

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
