## [Decision Letter · Decision Letter 0]

22 May 2023

PONE-D-23-03270HIV testing rates among the never-married young men in sub-Saharan Africa: an analysis of Demographic and Health Survey data (2010-2020)PLOS ONE

Dear Dr. Musonda,

Thank you for submitting your manuscript to PLOS ONE. After careful consideration, we feel that it has merit but does not fully meet PLOS ONE’s publication criteria as it currently stands. Therefore, we invite you to submit a revised version of the manuscript that addresses the points raised during the review process.

We look forward to receiving your revised manuscript.

Kind regards,

Ephraim Kumi Senkyire

Academic Editor

PLOS ONE

Journal Requirements:

2. Please upload a copy of Figure 1, to which you refer in your text on page 9. If the figure is no longer to be included as part of the submission please remove all reference to it within the text.

3.PLOS requires an ORCID iD for the corresponding author in Editorial Manager on papers submitted after December 6th, 2016. Please ensure that you have an ORCID iD and that it is validated in Editorial Manager. To do this, go to ‘Update my Information’ (in the upper left-hand corner of the main menu), and click on the Fetch/Validate link next to the ORCID field. This will take you to the ORCID site and allow you to create a new iD or authenticate a pre-existing iD in Editorial Manager. Please see the following video for instructions on linking an ORCID iD to your Editorial Manager account: https://www.youtube.com/watch?v=_xcclfuvtxQ.

Reviewers' comments:

Reviewer's Responses to Questions

**Comments to the Author**

1. Is the manuscript technically sound, and do the data support the conclusions?

Reviewer #1: Partly

Reviewer #2: Partly

2. Has the statistical analysis been performed appropriately and rigorously? 

Reviewer #1: Yes

Reviewer #2: Yes

3. Have the authors made all data underlying the findings in their manuscript fully available?

Reviewer #1: Yes

Reviewer #2: Yes

4. Is the manuscript presented in an intelligible fashion and written in standard English?

Reviewer #1: Yes

Reviewer #2: Yes

5. Review Comments to the Author

Reviewer #1: The work seeks to access the HIV testing rate among unmarried young men in SSA. It was well written. However, there are some concerns that you need to address

1. The aim of the study was not well stated.

2. Why was this study restricted to young unmarried men? Why did you exclude young unmarried women? Why did you also exclude married men and women? This was not explained in the manuscript.

3. How were the countries selected? Those excluded why were they excluded?

4. The discussion was poorly written. You did not relate it to other literatures on HIV testings among young unmarried men elsewhere. You did not give a possible reason for your findings and how it compared to findings elsewhere.

5. Though, you mentioned in your limitation section that the data were collected at different years and may have been collected earlier than published, how did you ensure that despite this disparity, the data could be compared without bias? Are you sure the situation has not changed in some of the countries?

Reviewer #2: The title does not read well and should be revised. The third paragraph of the discussion needs to be revised. Some of the information presented in the introduction and discussion seems assumptive and needs to be substantiated with literature. Additional data collection (other data sets) and analysis would be helpful in deriving more sound conclusions and suggested changes to policy that would be both realistic and sustainable. Acknowledgement of the range of data spanning a substantial time period (2010-2022) as a limitation was noteworthy but this does pose a problem to the accuracy of the analysis. Perhaps the data should be limited to a smaller time window as advances in HIV diagnosis, research and therapeutics have occurred at a rapid rate within the past 20 years. Furthermore, due to certain countries within SSA not have accessible DHS datasets this could possibly not be an accurate representation of SSA. This being said, the study does make use of good data analysis techniques and draws some important conclusions which may influence healthcare policies. Additional suggestions/recommendations could be helpful in this regard.

6. PLOS authors have the option to publish the peer review history of their article (what does this mean?). If published, this will include your full peer review and any attached files.

Reviewer #1: **Yes: **Tijani Idris Ahmad Oseni

Reviewer #2: No

---

## [Author Response · Author response to Decision Letter 0]

3 Jul 2023

The response letter of reviews has been attached.

---

## [Decision Letter · Decision Letter 1]

10 Jul 2023

PONE-D-23-03270R1Prevalence of HIV testing among the never-married young men (15-24) in sub-Saharan Africa: an analysis of Demographic and Health Survey data (2015-2022)PLOS ONE

Dear Dr. Musonda,

Thank you for submitting your manuscript to PLOS ONE. After careful consideration, we feel that it has merit but does not fully meet PLOS ONE’s publication criteria as it currently stands. Therefore, we invite you to submit a revised version of the manuscript that addresses the points raised during the review process.

We look forward to receiving your revised manuscript.

Kind regards,

Ephraim Kumi Senkyire

Academic Editor

PLOS ONE

Journal Requirements:

Additional Editor Comments:

Dear Author

for your manuscripts to be accepted, you need to provide point by point with reference to number line (s) in manuscript to reviewers comments . your Reponses to the reviewers can not be verified in the manuscript hence you need to indicate the number line(s) where to corrections were made

Reviewers' comments:

Reviewer's Responses to Questions

**Comments to the Author**

1. If the authors have adequately addressed your comments raised in a previous round of review and you feel that this manuscript is now acceptable for publication, you may indicate that here to bypass the “Comments to the Author” section, enter your conflict of interest statement in the “Confidential to Editor” section, and submit your "Accept" recommendation.

Reviewer #1: All comments have been addressed

2. Is the manuscript technically sound, and do the data support the conclusions?

Reviewer #1: Yes

3. Has the statistical analysis been performed appropriately and rigorously? 

Reviewer #1: Yes

4. Have the authors made all data underlying the findings in their manuscript fully available?

Reviewer #1: Yes

5. Is the manuscript presented in an intelligible fashion and written in standard English?

Reviewer #1: Yes

6. Review Comments to the Author

Reviewer #1: (No Response)

7. PLOS authors have the option to publish the peer review history of their article (what does this mean?). If published, this will include your full peer review and any attached files.

Reviewer #1: **Yes: **Tijani Oseni

---

## [Author Response · Author response to Decision Letter 1]

13 Jul 2023

Dear Editor,

Thank you for your review, we have attached the revised manuscript and rebuttal letter.

---

## [Editor Report · Decision Letter 2]

17 Jul 2023

PONE-D-23-03270R2Prevalence of HIV testing among the never-married young men (15-24) in sub-Saharan Africa: an analysis of Demographic and Health Survey data (2015-2022)PLOS ONE

Dear Dr. Musonda,

Thank you for submitting your manuscript to PLOS ONE. After careful consideration, we feel that it has merit but does not fully meet PLOS ONE’s publication criteria as it currently stands. Therefore, we invite you to submit a revised version of the manuscript that addresses the points raised during the review process.

My comment(s) were not addressed hence respond to it appropriately  before final decision can be made.

We look forward to receiving your revised manuscript.

Kind regards,

Ephraim Kumi Senkyire

Academic Editor

PLOS ONE

Journal Requirements:

Additional Editor Comments:

Hello

I asked in my previous message for you to address reviewers comment point by pint indicating the corresponding number line(s) however this was not done hence you need to to that.

---

## [Author Response · Author response to Decision Letter 2]

30 Jul 2023

Dear Editor,

Thank you for your guidance. We have updated the rebuttal letter to ensure that each of the reviewer's comments is addressed point by point indicating the corresponding number line(s) where the changes or edits have been made in the manuscript document We have also highlighted the said changes in the manuscript document. Looking forward to your further guidance.

Regards,

---

## [Editor Report · Decision Letter 3]

7 Aug 2023

PONE-D-23-03270R3Prevalence of HIV testing among the never-married young men (15-24) in sub-Saharan Africa: an analysis of Demographic and Health Survey data (2015-2022)PLOS ONE

Dear Dr. Musonda,

Thank you for submitting your manuscript to PLOS ONE. After careful consideration, we feel that it has merit but does not fully meet PLOS ONE’s publication criteria as it currently stands. Therefore, we invite you to submit a revised version of the manuscript that addresses the points raised during the review process.

Thank you for your responses however few corrections need to be done:

1. reference(s) is/are need for your reasons to include only unmarried men......

2.line 108: in your feedback to the reviewer, you indicted that a total of 19 countries were included however, in the manuscript you indicated 18 countries.  this discrepancy need to be resolved. 

We look forward to receiving your revised manuscript.

Kind regards,

Ephraim Kumi Senkyire

Academic Editor

PLOS ONE

Journal Requirements:

Additional Editor Comments:

Thank you for your responses however few corrections need to be done:

1. reference(s) is/are need for your reasons to include only unmarried men......

2.line 108: in your feedback to the reviewer, you indicted that a total of 19 countries were included however, in the manuscript you indicated 18 countries. this discrepancy need to be resolved.

---

## [Author Response · Author response to Decision Letter 3]

28 Aug 2023

Dear Editor,

Thank you for your guidance. We have updated the rebuttal letter to address the editors comments. We have also highlighted the said changes in the manuscript document. Looking forward to your further guidance.

Regards,

---

## [Editor Report · Decision Letter 4]

14 Sep 2023

Prevalence of HIV testing among the never-married young men (15-24) in sub-Saharan Africa: an analysis of Demographic and Health Survey data (2015-2020)

PONE-D-23-03270R4

Dear Dr. Musonda,

We’re pleased to inform you that your manuscript has been judged scientifically suitable for publication and will be formally accepted for publication once it meets all outstanding technical requirements.

Kind regards,

Ephraim Kumi Senkyire

Academic Editor

PLOS ONE
---

## [Editor Report · Acceptance letter]

28 Sep 2023

PONE-D-23-03270R4 

Prevalence of HIV testing uptake among the never-married young men (15-24) in sub-Saharan Africa: an analysis of Demographic and Health Survey data (2015-2020) 

Dear Dr. Musonda:

I'm pleased to inform you that your manuscript has been deemed suitable for publication in PLOS ONE. Congratulations! Your manuscript is now with our production department. 

Kind regards, 

on behalf of

Prof Ephraim Kumi Senkyire 

Academic Editor

PLOS ONE